

# An avian femur from the Late Cretaceous of Vega Island, Antarctic Peninsula: removing the record of cursorial landbirds from the Mesozoic of Antarctica

Abagael R. West[1,2,3], Christopher R. Torres[4], Judd A. Case[5], Julia A. Clarke[4,6], Patrick M. O'Connor[7,8] and Matthew C. Lamanna[1]

[1] Section of Vertebrate Paleontology, Carnegie Museum of Natural History, Pittsburgh, PA, USA
[2] Section of Mammals, Carnegie Museum of Natural History, Pittsburgh, PA, USA
[3] Department of Biological Sciences, University of Pittsburgh, Pittsburgh, PA, USA
[4] Department of Integrative Biology, University of Texas at Austin, Austin, TX, USA
[5] Department of Biology, Eastern Washington University, Cheney, WA, USA
[6] Jackson School of Geosciences, University of Texas at Austin, Austin, TX, USA
[7] Department of Biomedical Sciences, Ohio University Heritage College of Osteopathic Medicine, Athens, OH, USA
[8] Ohio Center for Ecology and Evolutionary Studies, Ohio University, Athens, OH, USA

Corresponding author
Abagael R. West,
westa@carnegiemnh.org

## ABSTRACT

In 2006, a partial avian femur (South Dakota School of Mines and Technology (SDSM) 78247) from the Upper Cretaceous (Maastrichtian) Sandwich Bluff Member of the López de Bertodano Formation of Sandwich Bluff on Vega Island of the northern Antarctic Peninsula was briefly reported as that of a cariamiform—a clade that includes extant and volant South American species and many extinct flightless and cursorial species. Although other authors have since rejected this taxonomic assignment, SDSM 78247 had never been the subject of a detailed description, hindering a definitive assessment of its affinities. Here we provide the first comprehensive description, illustration, and comparative study of this specimen. Comparison of characters that may be assessed in this femur with those of avian taxa scored in published character matrices refutes the inclusion of SDSM 78247 within Cariamiformes, instead supporting its assignment to a new, as-yet unnamed large-bodied species within the genus *Vegavis*, and therefore its referral to a clade of semiaquatic anseriforms. Important character states diagnostic of *Vegavis* + *Polarornis* include strong craniocaudal bowing of the femoral shaft, the presence of a distinct fossa just proximal to the fibular trochlea, and the broad and flat shape of the patellar sulcus. Referral to *Vegavis* is based on the presence of a distinctive proximocaudal fossa and distolateral scar. This genus was previously known only from *Vegavis iaai*, a smaller-bodied taxon from the same locality and stratigraphic unit. Our reassignment of SDSM 78247 to *Vegavis* sp. removes the record of cariamiform landbirds from the Antarctic Cretaceous.

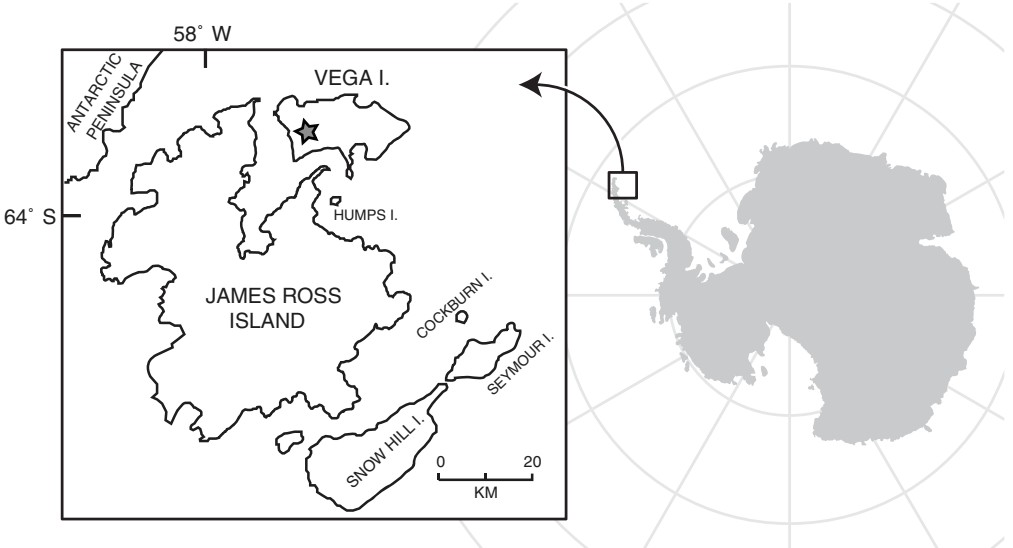

**Figure 1 Map of Vega Island, Antarctica, modified from *Roberts et al. (2014)*.** Star indicates approximate location of Sandwich Bluff, the site that yielded the avian femur (SDSM 78247) described herein.

## INTRODUCTION

In 2005, an expedition co-directed by one of us (J.A. Case) recovered a partial avian left femur as part of a field investigation of Upper Cretaceous sedimentary units in the James Ross Basin of the northern Antarctic Peninsula (Fig. 1). The femur (SDSM 78247; Figs. 2 and 3) was recovered from the Sandwich Bluff locality on the western half of Vega Island. The stratigraphic position of the specimen places it in the lower part of the uppermost Cretaceous (Maastrichtian) Sandwich Bluff Member of the López de Bertodano Formation, at a level some 12 m upsection from the concretionary horizon that yielded the holotype and referred skeletons of the anseriform bird *Vegavis iaai* (*Clarke et al., 2005*, *2016*).

A preliminary report of SDSM 78247 provisionally identified this femur as that of a terrestrial, cursorial bird belonging to either the otherwise exclusively Cenozoic Phorusrhacidae ('terror birds') or the extant Cariamidae (seriemas) within Cariamiformes (*Case et al., 2006*). This taxonomic assignment was based on the large size of the femur and three features of its distal morphology: an enlarged and caudally prominent tibiofibular crest, a laterally expansive lateral condyle, and a broad fibular trochlea. Nevertheless, in a review of Antarctic phorusrhacid fossils, *Cenizo (2012)* questioned this referral and instead identified a number of femoral character states that SDSM 78247 shares with various extant and Mesozoic foot-propelled diving birds (e.g., Hesperornithiformes, Gaviidae, Podicipedidae). These include a strongly curved shaft, a shallow patellar sulcus, a broad distal end with a laterally projecting fibular trochlea, a shallow intercondylar sulcus, a reduced fovea for insertion of the tendon of m. tibialis cranialis, and a long medial supracondylar crest. Subsequently, *Agnolín et al. (2017)* attributed

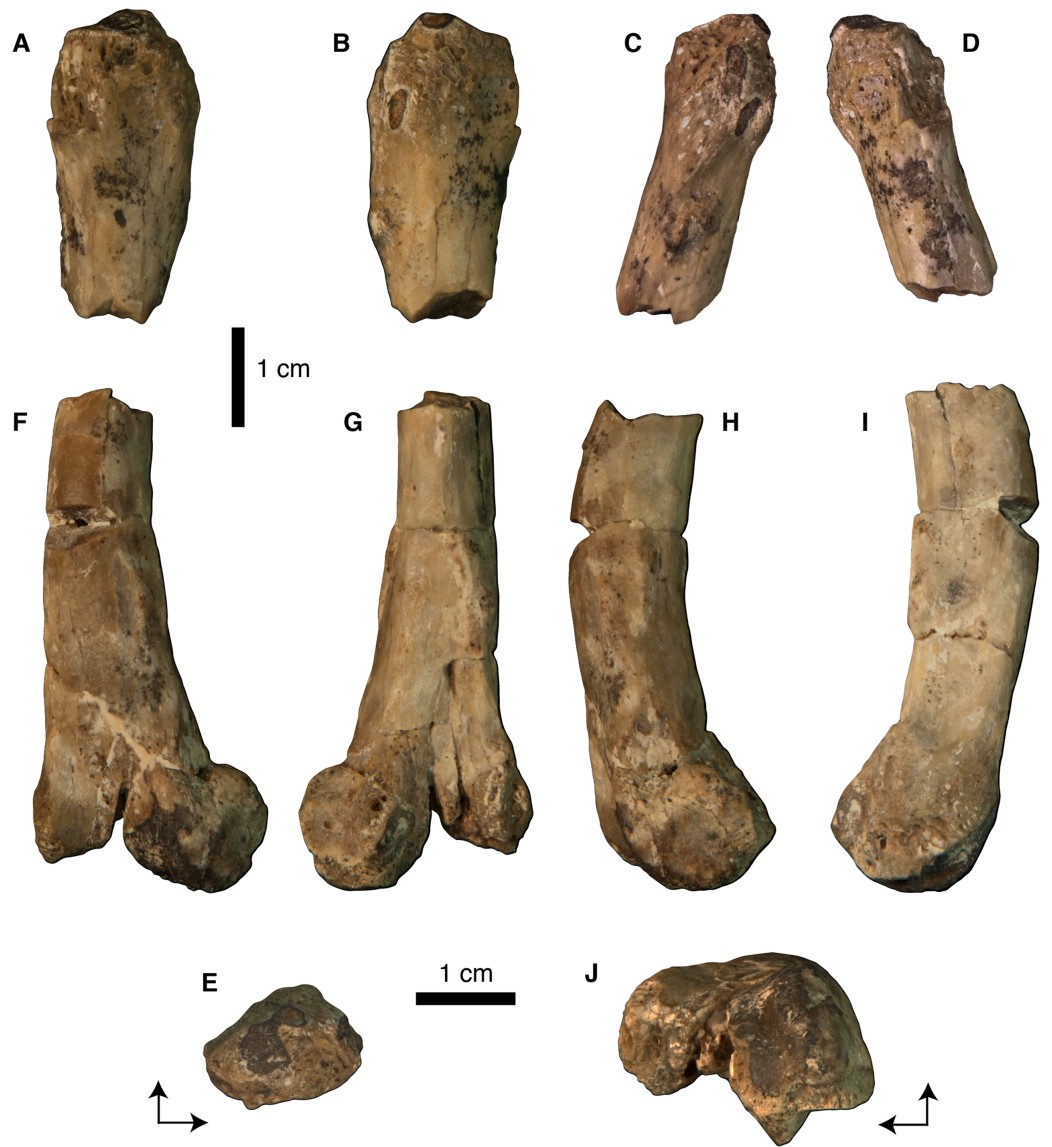

**Figure 2 Photographs of left femur of *Vegavis* sp. (SDSM 78247).** Proximal fragment in (A) cranial, (B) caudal, (C) lateral, (D) medial, and (E) proximal views; distal fragment in (F) cranial, (G) caudal, (H) lateral, (I) medial, and (J) distal views. Arrows in E and J indicate cranial (top of image) and medial (left side of image) directions.

SDSM 78247 to an indeterminate taxon within Vegaviidae, their newly proposed clade of Gondwanan neognathous waterbirds that diversified in the Late Cretaceous and allegedly survived into the Paleocene. *Agnolín et al. (2017)* regarded SDSM 78247 as a member of Vegaviidae based on a suite of features that the specimen shares with the putative vegaviids *Polarornis gregorii* and *V. iaai*. *Agnolín et al.'s (2017)* referral of several other Cretaceous and Paleogene taxa to Vegaviidae has recently been contested (*Mayr et al., 2018*). Here we provide the first detailed description and systematic comparison of SDSM 78247.

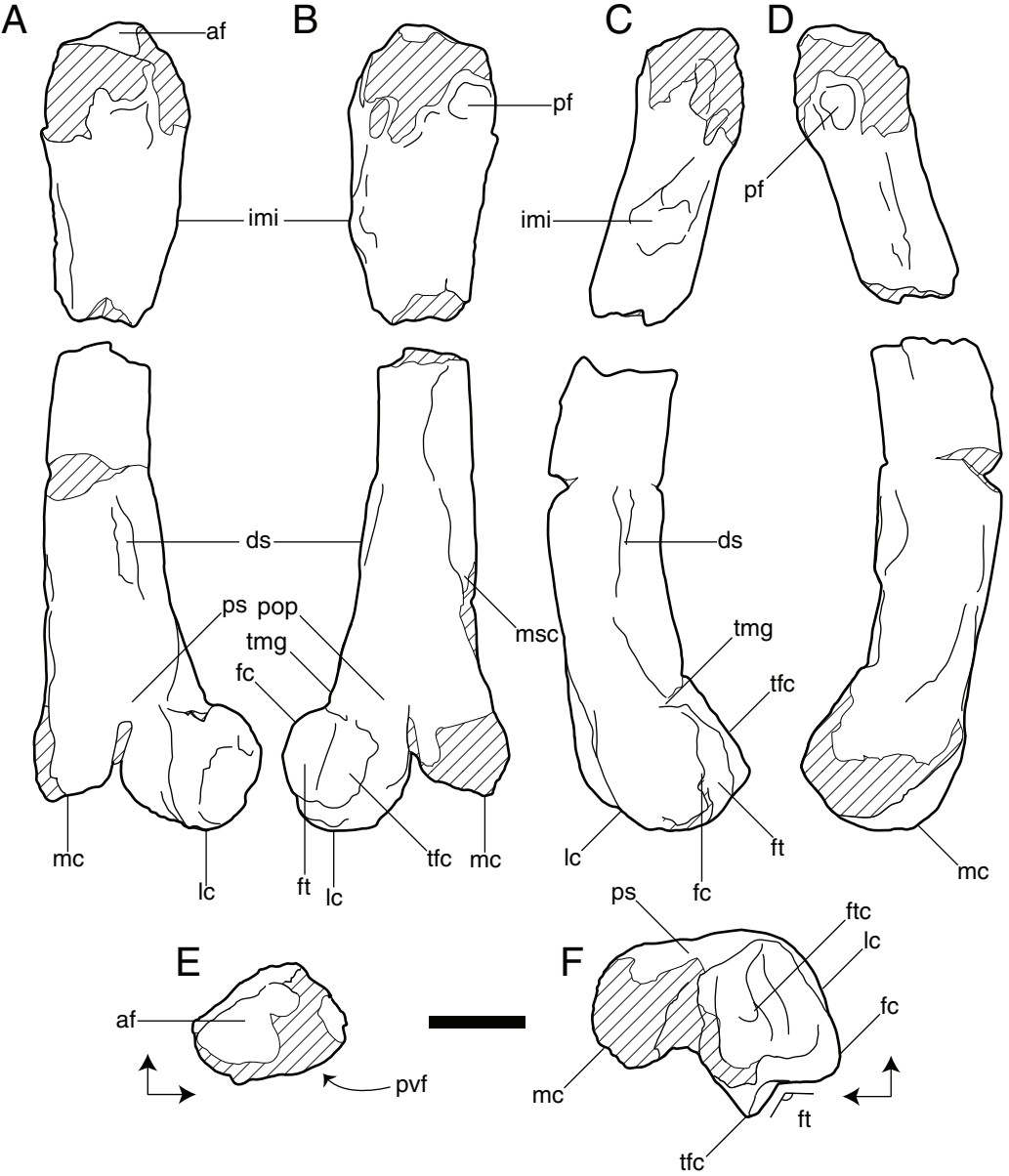

**Figure 3 Line drawings of left femur of *Vegavis* sp. (SDSM 78247).** (A) Cranial, (B) caudal, (C) lateral, (D) medial, (E) proximal, and (F) distal views. Arrows in E and F indicate cranial (top of image) and medial directions. Hatching indicates worn or broken areas. Abbreviations: af, antitrochanteric face; ds, distolateral scar; fc, fibular crest; ft, fibular trochlea; ftc, fovea for insertion of tendon of m. tibialis cranialis; imi, insertion for m. iliotrochantericus; lc, lateral condyle; mc, medial condyle; msc, medial supracondylar crest; pf, proximocaudal fossa; pop, popliteal fossa; ps, patellar sulcus; tfc, tibiofibular crest; tmg, tubercle for insertion of m. gastrocnemialis. Scale bar = 1 cm.

## SYSTEMATIC PALEONTOLOGY

Aves Linnaeus, 1758

Galloanserae Sibley, Ahlquist, and Monroe, 1988

Anseriformes Wagler, 1831

*Vegavis* Clarke, Tambussi, Noriega, Erickson, and Ketcham, 2004
Species indeterminate

### Referred specimen
SDSM 78247, a partial left femur preserved in two pieces.

### Locality and horizon
Locality V2005-3, 'Plesiosaur Papoose,' Sandwich Bluff, Vega Island, Antarctic Peninsula. Lower part (≈Unit SBM2 of *Roberts et al. (2014)*; J.A. Case, 2018, personal observations) of the Upper Cretaceous (Maastrichtian) Sandwich Bluff Member of the López de Bertodano Formation. This location is about 48 m below the stratum that yielded the hadrosaur tooth described by *Case et al. (2000)*. See *Roberts et al. (2014)* for detailed stratigraphic information on the Sandwich Bluff site.

## DESCRIPTION
SDSM 78247 is a largely complete left femur (Figs. 2 and 3) that is preserved in two pieces; both ends are present but the middle portion of the shaft is missing. The proximal piece is 3.2 cm in length and 1.4 cm wide across the proximalmost portion of the shaft. The distal piece is 4.7 cm in length and 2.2 cm wide across the distal condyles. Though the femoral head and most of the femoral trochanter are not preserved, the latter was probably craniocaudally narrow as estimated from its broken base. On the caudal surface of the proximal fragment, near the proximal end and immediately lateral to the broken base of the femoral head, there is a well-defined, circular proximocaudal fossa (Figs. 3B and 3D). The insertion of the m. iliotrochantericus takes the form of a prominent quadrangular tubercle on the lateral side of the proximal end of the shaft (Figs. 3A–3C). Overall, the proximal fragment of the femur is wider mediolaterally than deep craniocaudally, though these dimensions become progressively more equal distally such that the broken distal end of the proximal fragment is subcircular in cross section. Although a portion of the shaft is missing, the femur was clearly markedly bowed cranially.

The distal portion of SDSM 78247 is laterally compressed (i.e., deeper craniocaudally than wide mediolaterally) and oval in cross section at its broken proximal end. The relative bone wall thickness (RBT, sensu *Smith & Clarke, 2014*) is approximately 36%. There is a subtle distolateral scar located on the lateral face of the shaft at the approximate midlength of the distal piece (i.e., roughly three-fourths the estimated length of the femur from its proximal end; Figs. 3A–3C). At approximately the same proximodistal level, the medial supracondylar crest expands into a massive, proximodistally elongate tuberosity that projects caudally from the remainder of the femoral shaft (Fig. 3B). Distally, there is a broad, shallow patellar sulcus on the cranial surface of the bone, and the region of the intercondylar sulcus is poorly preserved (Figs. 3A and 3F). The medial condyle is damaged, inhibiting comparisons with the lateral condyle. On the lateral condyle, the fibular trochlea is broad, laterally expansive, and strongly proximally deflected (Figs. 3B, 3C and 3F). Its lateral and medial margins are formed by the prominent but slightly weathered fibular and tibiofibular crests, respectively, with a broad, shallow, proximocranially directed

groove occupying the space between these crests (Figs. 3B, 3C and 3F). There is a low medial epicondyle proximal to the medial condyle (Figs. 3A, 3B and 3D).

## COMPARISONS

SDSM 78247 and *Vegavis iaai* share two features that were proposed by *Clarke et al. (2016)* to differentiate *Vegavis* from *Polarornis*. The first of these is the presence of a deep, round ligament scar on the proximocaudal face. The new femur exhibits a condition similar to that of *V. iaai*: in both, this scar forms a round fossa with a distinct lip around at least the proximolateral margin (Figs. 4I and J). By contrast, this scar is shallow and poorly defined in anatids (e.g., *Mergus serrator*, *Anas platyrhynchos*, and *Anser anser*) and absent in screamers (e.g., *Chauna torquata*) and galliforms (e.g., *Gallus gallus* and *Meleagris gallopavo*). The scar is present as a raised structure in other foot-propelled diving taxa such as loons (e.g., *Gavia stellata*) and grebes (e.g., *Aechmophorus occidentalis* and *Podilymbus podiceps*).

The new femur and *V. iaai* also share an elongate scar on the distolateral margin, a feature that is not observed in the holotypic specimen of *Polarornis gregorii* (Figs. 4A–C, 4E–G and 4I–K). A similar distolateral scar has been reported in the distal femur of a partial skeleton from the López de Bertodano Formation of Seymour Island that has been tentatively assigned to *Polarornis* (MLP 96-I-6-2; *Acosta Hospitaleche & Gelfo, 2015*; *Agnolín et al., 2017*; *Mayr et al., 2018*); however, this bone is poorly preserved, and we were unable to assess the presence or form of this scar as it was originally figured. The presence, form, and position of this distolateral scar are highly variable across Neognathae. In *V. iaai* and SDSM 78247, the scar forms a single, low ridge that is positioned near the cranial margin in lateral view and well proximal to the lateral condyle. Among anatids, this scar abuts the lateral condyle near the caudal margin in *A. platyrhynchos* and *A. anser*; in *M. serrator*, by contrast, it occupies a position similar to that seen in *V. iaai* but is bipartite, forming two distinct scars. A scar in this location is not observed in screamers (e.g., *C. torquata*) or galliforms (e.g., *G. gallus*, *M. gallopavo*). In other foot-propelled diving birds, this scar is present as two widely spaced tubercula either near the caudal margin (in crown-group loons) or near the midline (in grebes). However, a conspicuous raised scar in this location is absent in proposed stem loons (e.g., *Colymboides minutus*; *Storer, 1956*).

SDSM 78247 can be easily differentiated from the holotype and referred specimens of *V. iaai* (*Clarke et al., 2005*, *2016*) by several features, including absolute size; the former is nearly twice the size of the femora of the latter two skeletons. Also, in SDSM 78247, the intersection between the lateral margin of the femoral shaft and the articular surface of the fibular trochlea appears abrupt rather than gradational in caudal view. In the new femur, the proximal part of the trochlea is rotated proximally, generally a notch-like intersection (*Worthy et al., 2017*: character 220) (Fig. 4I). By contrast, in *V. iaai*, as well as in *P. gregorii*, the proximal margin of the fibular trochlea grades smoothly into the shaft (Figs. 4J and 4K). Additionally, the round proximocaudal ligament scar diagnostic of *Vegavis* is close to the lateral margin in *V. iaai* (Fig. 4J) but positioned slightly nearer to the midline in SDSM 78247 (Fig. 4I). Furthermore, in *V. iaai*, the margin of this fossa is

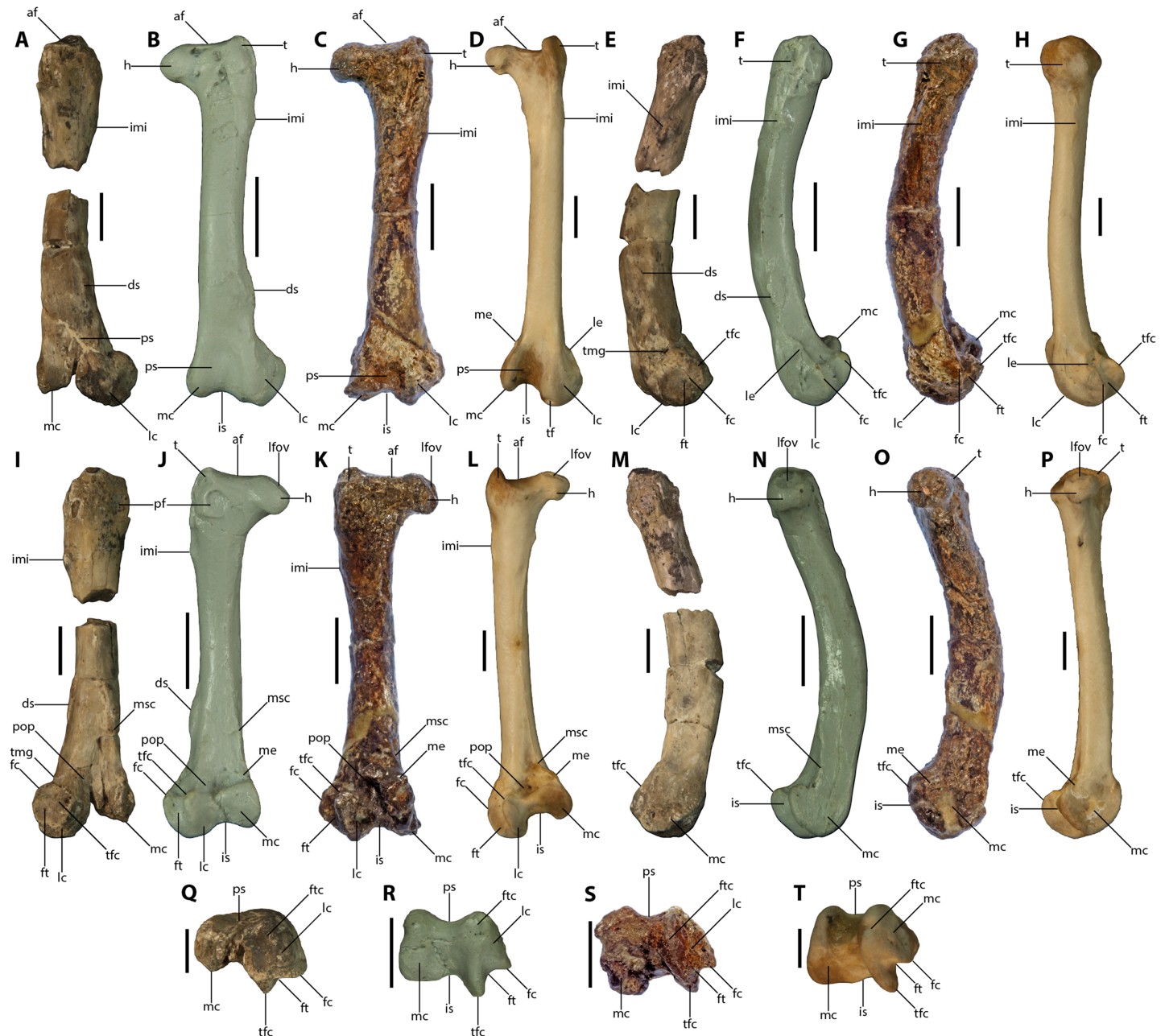

**Figure 4 Comparison of left femora.** (A, E, I, M, Q) *Vegavis* sp. (SDSM 78247), (B, F, J, N, R) *Vegavis iaai* (cast of MACN-PV 19.148, reversed), (C, G, K, O, S) *Polarornis gregorii* (TTU P 9265), and (D, H, L, P, T) *Cariama cristata* (TMM M-10446), in cranial (A–D), lateral (E–H), caudal (I–L), medial (M–P), and distal (Q–T) views. Abbreviations as in Fig. 3 except: h, femoral head; is, intercondylar sulcus; le, lateral epicondyle; me, medial epicondyle; t, femoral trochanter. Scale bar = 1 cm.

circumscribed by a prominent lip, though this lip is least conspicuous medially. In the new femur, however, only the proximolateral margin of this fossa is bordered by a lip. The raised distolateral scar characteristic of *Vegavis* is better developed in *V. iaai* than SDSM 78247 (Figs. 4A, 4B, 4E, 4F, 4I and 4J), though this may be due to weathering of the latter. In distal view, the medial condyle of SDSM 78247 appears proportionally slightly smaller relative to the lateral condyle than in *V. iaai*, however, this area is heavily damaged

in SDSM 78247 (Figs. 4Q and 4R). Also in distal view, the fibular trochlea of SDSM 78247 appears slightly shallower and more laterally rotated than that of *V. iaai* (Figs. 4Q and 4R). Finally, the bone wall of the new femur is proportionally thicker (RBT ≈ 36%) than in *V. iaai* (RBT ≈ 21.6%; *Garcia Marsà, Agnolín & Novas, 2017*).

Of the 290 characters in the phylogenetic data matrix of *Worthy et al. (2017)*, 35 pertain to the femur, and 21 of these (i.e., 7% of the total character set) could be definitively scored in SDSM 78247. SDSM 78247 was scored identically to both *V. iaai* and *P. gregorii* (as both of these species were scored by *Agnolín et al., 2017*) for 19 of the 21 femoral characters in the *Worthy et al. (2017)* matrix that could be assessed in the new fossil. These characters include: (188) a concave antitrochanteric articular face; (193) a lack of pneumatic openings on the caudal face adjacent to the antitrochanteric articular face; (194) the conformation of the obturator impression as a single large scar near the antitrochanteric articular face; (195) a weakly developed scar for insertion of the medial part of the m. puboischiofemoralis; (198) a weakly marked impression for the m. iliofemoralis internus; (199) an elongate shaft with subparallel medial and lateral margins; (200) a strongly craniocaudally bowed shaft; (201) a relatively straight medial face in caudal view; (202) the position of the scar for insertion of the m. iliotrochantericus caudalis at mid-craniocaudal depth on the lateral face; (204) the position of the tuberosity of the medial crest at the medial margin of the caudal face; (206) widely separated insertions for the m. obturatorius lateralis and the m. ischiofemoralis; (207) the m. gastrocnemialis lateralis tubercle forms a rugose scar on the lateral face proximocranial to the fibular trochlea; (211) the medial condyle comprises approximately half of the total width across the condyles; (213) the patellar sulcus is broad and flat in cranial view; (214) the presence of a notch for the m. tibialis tendon on the distal end of the lateral condyle; (215) a shallow popliteal fossa; (218) the medial supracondylar crest is short with a notched medial profile; (219) the fibular trochlea and lateral condyle extend equally distally; and (220) the proximal part of the articular surface of the fibular trochlea is rotated cranially, forming a prominence that is markedly offset from the lateral face. Additionally, although the trochanteric crest in SDSM 78247 is missing, it was likely craniocaudally narrow as in *V. iaai* and *P. gregorii*. Characters from the *Worthy et al. (2017)* matrix that are apomorphic for SDSM 78247 are (205) orientation of the lateral condyle in cranial view divergent from axis of the shaft; and (209) the absence or reduction of a distinct depression on the caudal face immediately proximal to the fibular trochlea (see Supplemental File for character scorings).

Although SDSM 78247 was originally assigned to Cariamiformes (*Case et al., 2006*), it differs markedly from the femora of all members of this clade; for example, no taxon within Cariamiformes possesses the distinct scars discussed above. The new femur also differs from *Cariama cristata* with respect to 11 of the 21 scorable characters from the *Worthy et al. (2017)* matrix, including the following states that are present in *C. cristata* but not in the new specimen: (194) the conformation of the obturator impression as two scars; (195) the presence of a strongly developed scar for the insertion of the m. puboischiofemoralis pars medialis; (198) the impression for the m. iliofemoralis internus is a well-marked rugosity; (200) a shaft that is straight in lateral view; (207) the presence of the tubercle for the m. gastrocnemialis lateralis as a round scar near the

fibular trochlea; (209) the absence of a distinct depression immediately proximal to the caudal articular surface of the fibular trochlea; (211) the medial condyle contributing more than half of the maximum mediolateral width across the condyles; (215) a deep popliteal fossa; (218) the lateral edge of the distal end of the shaft in caudal view is smoothly curving and continuous with the condyle; and (220) the fibular trochlea is caudally directed, merging smoothly into the shaft. Additionally, the circular proximocaudal fossa present in the new femur is absent in both *C. cristata* (which is convex in this area, lacking any depressions) and phorusrhacids (*Alvarenga & Hofling, 2003*). The distolateral scar present in SDSM 78247 is also absent in both *C. cristata* and phorusrhacids (*Alvarenga & Hofling, 2003*). Furthermore, in Cariamiformes, the patellar sulcus is deep and is bordered by sharp crests on the cranial parts of the condyles (*Alvarenga & Hofling, 2003*). By contrast, in the new femur, the patellar sulcus is shallow and broad.

## DISCUSSION

The morphology of SDSM 78247 is consistent with its membership in *Vegavis*. This assertion is based on the identical scores of the new femur and *V. iaai* with respect to 19 of the 21 scorable characters from the phylogenetic data matrix of *Worthy et al. (2017)*. SDSM 78247 also exhibits the circular proximocaudal fossa that is synapomorphic of *Vegavis*, as well as the elongate distolateral scar that is diagnostic of *Vegavis* but not *Polarornis*. SDSM 78247 may be differentiated from *V. iaai* based on aspects of the distolateral femur, as well the overall size of the element and the position and shape of the proximocaudal fossa. These distinctions suggest that the new specimen likely represents a new species of *Vegavis*; however, this putative new form is, at present, too incompletely represented to warrant formally erecting a new taxon.

Of the four femoral character states proposed by *Agnolín et al. (2017)* as diagnostic for Vegaviidae that are scorable in SDSM 78247, the new femur is consistent with three: (1) craniocaudal bowing of the shaft; (2) the presence of a distinct fossa just proximal to the fibular trochlea; and (3) a broad and flat shape of the patellar sulcus. SDSM 78247 is inconsistent with the fourth proposed state, the presence of the obturator impressions as two separate, rugose scars. However, *V. iaai* is also inconsistent with this state (sensu *Worthy et al. (2017)* but contra *Agnolín et al. (2017)*). Thus, this conformation of the obturator impressions is likely not diagnostic of *Vegavis* + *Polarornis*.

Our reassessment of the partial avian femur SDSM 78247 reveals the presence of a comparatively large-bodied and likely new *Vegavis* species from the latest Cretaceous of Vega Island, Antarctica, and unambiguously removes the only record of a cursorial bird (specifically Cariamiformes; *Case et al., 2006*) from the Mesozoic of that continent. Morphological comparisons presented herein ally SDSM 78247 more closely with *V. iaai* than with any other sampled taxon, and as such are consistent with the placement of the specimen within *Vegavis*. SDSM 78247 is distinguished from *V. iaai* and *P. gregorii* both by morphology and overall size; the new femur is most similar in size to, though still larger than, that of *Polarornis* sp. MLP 96-I-6-2 (*Acosta Hospitaleche & Gelfo, 2015*).

The osteohistology of all previously described *Vegavis* (MLP 93-I-3-1, MACN-PV 19.748) and *Polarornis* (TTU P 9265) specimens indicates that these birds had reached adulthood, or nearly so, at the time of death (*Chinsamy, Martin & Dodson, 1998*; *Clarke et al., 2005*, *2016*; *Garcia Marsà, Agnolín & Novas, 2017*). As such, the much greater size of SDSM 78247 cannot be easily explained by ontogenetic variability, and instead suggests taxonomic distinction. The new femur exhibits *Vegavis*-like states of some characters that have been proposed to differentiate that genus from *Polarornis* (i.e., proximocaudal fossa, distolateral scar; *Clarke et al., 2016*; *Agnolín et al., 2017*; *Mayr et al., 2018*). SDSM 78247 also has a much greater relative bone thickness than *V. iaai*, corresponding more closely to *P. gregorii* (RBT = 37%; *Chinsamy, Martin & Dodson, 1998*; *de Mendoza & Tambussi, 2015*) in this regard, but this may be a consequence of allometric considerations and/or functional scaling in aquatic/diving-specialized taxa. Extensive long bone osteosclerosis such as that observed in SDSM 78247 has been proposed as a correlate of diving specialization, and it scales with positive allometry (e.g., *Chinsamy, Martin & Dodson, 1998*; *de Mendoza & Tambussi, 2015*). The identification of SDSM 78247 as belonging to a previously unrecognized species within *Vegavis* substantially increases the known body size range of this taxon.

## CONCLUSIONS

Our restudy of the isolated avian femur SDSM 78247 from the Upper Cretaceous López de Bertodano Formation of Antarctica—which was previously attributed to Cariamiformes, constituting the lone Cretaceous record of that clade—demonstrates that the specimen instead pertains to a close relative of the waterbird *Vegavis iaai*, known from the same locality and stratigraphic unit. SDSM 78247 bears a strong morphological resemblance to the femur of *V. iaai* but is much larger than both this taxon and *Polarornis gregorii*, which is known from deposits just prior to the Cretaceous/Paleogene boundary on nearby Seymour Island. The refutation of SDSM 78247 as a member of Cariamiformes has important paleobiogeographical implications. The geographical connectivity and the timing of exchange between faunas from South America and other parts of the globe during the Mesozoic and Cenozoic remain contentious (e.g., *Mayr, 2009*). However, this proposed cariamiform record from the Cretaceous of Antarctica—which was formerly the most ancient, globally—no longer requires explanation. The earliest records of Cariamiformes instead consist of forms from the Paleogene of Europe and a tentatively referred specimen from South America (*Mayr, 2009*; *Mayr, Alvarenga & Clarke, 2011*).

## INSTITUTIONAL ABBREVIATIONS

**MACN**   Museo Argentino de Ciencias Naturales, Buenos Aires, Argentina
**MLP**   Museo de La Plata, La Plata, Argentina
**SDSM**   South Dakota School of Mines and Technology, Rapid City, SD, USA
**TMM**   Texas Memorial Museum, Austin, TX, USA
**TTU**   Museum of Texas Tech University, Lubbock, TX, USA

## ACKNOWLEDGEMENTS

This research began as one of a series of investigations led by one of us (J.A. Case) in concert with researchers from the MLP and the Instituto Antártico Argentino. We thank Dr. James Martin of the School of Geosciences at the University of Louisiana at Lafayette for his collection (on February 18, 2005) and curation of the specimen which has been studied here.

We also thank D. Edward Malinzak (Black Hills State University) for locating SDSM 78247 in the SDSM Museum of Geology collections, and Sally Shelton and Darrin Pagnac (SDSM) for loan of the specimen (to M.C.L.). We are grateful to Mary Hennen and John Bates (Field Museum of Natural History; FMNH) for photographing the femora of *Phoebastria nigripes* (FMNH 313761) and *Macronectes halli* (FMNH 339546). We thank Federico Agnolín (MACN) for sharing his original photographs of *Polarornis gregorii* (TTU P 9265) and providing helpful review comments on the manuscript, and Andrew Farke (Raymond M. Alf Museum of Paleontology) and Gerald Mayr (Senckenberg Research Institute) for additional helpful review comments and advice.

### Funding

Support was provided by the Rea Postdoctoral Fellowship of Carnegie Museum of Natural History (to Abagael R. West) and by the National Science Foundation Office of Polar Programs grants 0003844 (to Judd A. Case), 0087972 (to James Martin), ANT-1142129 (to Matthew C. Lamanna), ANT-1141820 (to Julia A. Clarke), and ANT-1142104 (to Patrick M. O'Connor). The funders had no role in study design, data collection and analysis, decision to publish, or preparation of the manuscript.

### Grant Disclosures

The following grant information was disclosed by the authors:
Rea Postdoctoral Fellowship of Carnegie Museum of Natural History.
National Science Foundation Office of Polar Programs grants: 0003844, 0087972, ANT-1142129, ANT-1141820, and ANT-1142104.

### Competing Interests

Julia A. Clarke is an Academic Editor for *PeerJ*.

### Author Contributions

- Abagael R. West conceived and designed the experiments, performed the experiments, analyzed the data, prepared figures and/or tables, authored or reviewed drafts of the paper, approved the final draft.
- Christopher R. Torres conceived and designed the experiments, performed the experiments, analyzed the data, prepared figures and/or tables, authored or reviewed drafts of the paper, approved the final draft.
- Judd A. Case contributed reagents/materials/analysis tools, authored or reviewed drafts of the paper, approved the final draft.

- Julia A. Clarke contributed reagents/materials/analysis tools, authored or reviewed drafts of the paper, approved the final draft.
- Patrick M. O'Connor authored or reviewed drafts of the paper, approved the final draft.
- Matthew C. Lamanna conceived and designed the experiments, contributed reagents/materials/analysis tools, authored or reviewed drafts of the paper, approved the final draft.

## Field Study Permissions

The following information was supplied relating to field study approvals (i.e., approving body and any reference numbers):

Fieldwork was conducted under the auspices of the National Science Foundation Office of Polar Programs.

## Data Availability

The new specimen is accessioned as SDSM 78247 at the Museum of Geology, South Dakota School of Mines and Technology. The raw data are available in the figures with scale bars and the Supplemental File (new character scorings).

## Supplemental Information

Supplemental information for this article can be found online at http://dx.doi.org/10.7717/peerj.7231#supplemental-information.

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
