# Peer review of "An avian femur from the Late Cretaceous of Vega Island, Antarctic Peninsula: removing the record of cursorial landbirds from the Mesozoic of Antarctica"

_PeerJ, doi:10.7717/peerj.7231_

## Round 0.1 · original submission · Minor Revisions

Overall, this manuscript is concisely written and its conclusions are well supported by the underlying data. The reviewers and I have identified a few minor areas for revision, which are included in the editorial notes below and the reviews (including marked-up PDFs).

- Please revise the abstract to include specific results. As is, it presents the overall conclusions, but some of the specific data underlying these conclusions in terms of characters, etc., should also be mentioned.

- lines 63-69 -- consider mentioning (briefly) which characters/data were used to support these assignments/reassignments

·

Basic reporting

This is a short note, which corrects the previous identification of an avian femur from the Latre Cretaceous of Antarctica, and for the first time presents a description of the fossil. As such, the manuscript is certainly a worthwile contribution to palaeornithology and I have only a few comments the authors may wish to take into account.
Technical details, such as the approariate use of the references, English grammar, etc. do not need to be improved except for the few minor instances listed below.

Experimental design

The paper follows current standards in palaeontological studies, but I wonder whether there should be a material and method section, in which institutional abbreviations are given and the used terminology is referenced. Certainly, this is not mandatory given the expected briefness of such a section and if it conforms with the journal standards, a material and method section can be omitted.

Validity of the findings

I full agree with the referal of the fossil to the Vegaviidae and have only a few minor comments the authors may wish to take into account (see below)

Additional comments

While I fully agree with the referral of this fossil to the Vegaviidae, the distal view of the distal end shows some differences to the distal femur of Vegavis in the depth of the patellar sulcus and the shape of the remaining lateral condyle (Fig. 4P, Q). I assume that these are due to the poor preservation of the fossil and it may be worth to add a specific note on this to the next (you already included some similar statements in lines 154ff).

You compare the new fossil with the holotypes of vegavis and Polarornis. What I miss, however, are comaprisons with the Vegavis/Polaronis material listed by Acista Hospitaleche in her paper on the putative gaviiforms from Antarctica. It would at least be necessary to state whether the new femur is also larger than the remains reported in this latter study.

Some minor comments:

- note that in avian terminology of the femur, the directions "cranial/caudal" rather than "dorsal/ventral" are used
- line 105: "is approximately 36" - add "mm" here
- line 152: "Additionally, the round proximoventral ligament scar of Vegavis is positioned near the midline in the new femur ... but is closer to the lateral margin in V. iaai." - as kit is, this sentence makes no sense, since you compare "Vegavis" with "Vegavis iaai". Is the new fossil meant here? In this case, it would be better to give the specimen number.
- caption of Fig. 1: "site" instead of "si te"
- caption of Fig. 3: "tendineus m. tibialis cranialis" - this should probably read "tendon of..."

As a side notes: Personally, I am not fully convinced that it is necessary and appropriate to keep Vegavis and Polarornis as two different taxa on the "genus"-level and I am not sure whether the differences between Vegavis and Polarornis are not the result of the poor preservation of the holotype of P. gregorii (e.g., concerning the absence of the distolateral scar). This question (synonymy of Vegavis and Polarornis) is certainly beyond the scope of the present manuscript, but it may ultimately bear on the affinities of the fossil within Vegaviidae, i.e., whether it is more closely related to Vegavis or Polarornis.

(Gerald Mayr)

·

Basic reporting

No comment

Experimental design

No comment

Validity of the findings

No comment

Additional comments

Dear Editor,

The article is concise and well-written. It includes novel information about a very interesting and potentially important specimen. I congratulate authors for such interesting contribution.
Some minor comments are included in the attached PDF.
I think the article should be accepted after minor corrections.

All the best,

Federico Agnolin

---

## Round 0.2 · accepted · Accept

Thank you for your close attention to the reviewer and editorial comments!